# Interaction between Non-Coding RNAs and Androgen Receptor with an Especial Focus on Prostate Cancer

**DOI:** 10.3390/cells10113198

**Published:** 2021-11-16

**Authors:** Mohammad Taheri, Tayyebeh Khoshbakht, Elena Jamali, Julia Kallenbach, Soudeh Ghafouri-Fard, Aria Baniahmad

**Affiliations:** 1Skull Base Research Center, Loghman Hakim Hospital, Shahid Beheshti University of Medical Sciences, Tehran 1983535511, Iran; Mohammad_823@yahoo.com; 2Institute of Human Genetics, Jena University Hospital, 07747 Jena, Germany; julia.kallenbach@uni-jena.de; 3Phytochemistry Research Center, Shahid Beheshti University of Medical Sciences, Tehran 1983535511, Iran; sare.khoshbakht@gmail.com; 4Department of Pathology, Loghman Hakim Hospital, Shahid Beheshti University of Medical Sciences, Tehran 1983535511, Iran; elena.jamali@yahoo.com; 5Department of Medical Genetics, School of Medicine, Shahid Beheshti University of Medical Sciences, Tehran 1983535511, Iran

**Keywords:** androgen receptor, miRNA, lncRNA, circular RNAs, prostate cancer

## Abstract

The androgen receptor (AR) is a member of the nuclear receptor superfamily and has three functional domains, namely the N-terminal, DNA binding, and C-terminal domain. The N-terminal domain harbors potent transactivation functions, whereas the C-terminal domain binds to androgens and antiandrogens used to treat prostate cancer. AR has genomic activity being DNA binding-dependent or through interaction with other DNA-bound transcription factors, as well as a number of non-genomic, non-canonical functions, such as the activation of the ERK, AKT, and MAPK pathways. A bulk of evidence indicates that non-coding RNAs have functional interactions with AR. This type of interaction is implicated in the pathogenesis of human malignancies, particularly prostate cancer. In the current review, we summarize the available data on the role of microRNAs, long non-coding RNAs, and circular RNAs on the expression of AR and modulation of AR signaling, as well as the effects of AR on their expression. Recognition of the complicated interaction between non-coding RNAs and AR has practical importance in the design of novel treatment options, as well as modulation of response to conventional therapeutics.

## 1. Introduction

The androgen receptor (AR), alternatively named as NR3C4 (nuclear receptor subfamily 3, group C, member 4), is a nuclear receptor [1] that is activated by a number of androgens, such as testosterone and its more active form, dihydrotestosterone [2]. AR has three protein domains, namely the N-terminal transcriptional regulation, DNA binding, and C-terminal ligand binding domain [3]. After being activated by its ligands in the cytoplasm, AR is transferred into the nucleus, where it exerts its main DNA-binding-dependent functions [4]. In fact, AR is a cytoplasmic protein in the absence of its ligands associated with chaperone proteins, such as heat shock proteins and co-chaperones. Androgen binding to the AR results in conformational changes, dissociating it from chaperone proteins [4]. After translocation to the nucleus, the androgen/AR complex dimerizes and binds to androgen response elements (AREs), which are present in the AR target genes. This process is involved in the regulation of the expression of target genes [5]. In addition to this canonical route of action, the androgen/AR complex has other functions that are mediated through non-DNA-binding-dependent routes [6]. Modulation of activity of ERK, AKT, and MAPK pathways are examples of this kind of function [6,7,8].

Notably, specific coregulators have been found to modulate the transcriptional activity of androgen/AR complex. The binding of coregulators with androgen/AR complex can either enhance (via coactivators) or suppress (via corepressors) its transactivation capability. This process is accomplished via the epigenetic changing of chromatin configuration and histone modifications [5].

A bulk of evidence indicates that non-coding RNAs (ncRNAs) have functional interactions with AR. This type of interaction is implicated in the pathogenesis of human malignancies, particularly prostate cancer. In the current review, we summarize the available data on the role of microRNAs (miRNAs), long non-coding RNAs (lncRNAs), and circular RNAs (circRNAs) on expression of AR, as well as the effects of AR on their expression.

## 2. Effects of miRNAs on AR

### 2.1. Acting on AR mRNA to Directly Negatively Regulate AR Expression

Several miRNAs have been found to suppress the expression of AR. In fact, most of these miRNAs have been shown to bind with 3′UTR of AR transcript, thus inducing its degradation or translation suppression. The interactions between miRNAs and AR mRNA, mainly through binding with its 3′UTR, have been mostly assessed in the context of prostate cancer. MiR-299-3p is one of the AR-interacting miRNAs. Expression of miR-299-3p has been reported to be decreased in prostate cancer samples, compared to noncancerous prostate samples. The restoration of miR-299-3p in prostate cancer cells has led to the induction of cell cycle arrest, reduction of cell proliferation and migration, and enhancement of levels of apoptotic markers. Moreover, the up-regulation of miR-299-3p leads to reduced expressions of AR, PSA, and VEGFA, suppressed epithelial mesenchymal transition (EMT), reduced levels of Slug, TGF-β3, p-AKT, and p-PRAS40, and enhanced E-cadherin levels. Taken together, miR-299-3p exerts anti-tumor effects via affecting activity of AR and VEGFA pathways [9].

Another study in prostate cancer cells has identified miR-185-binding sites in the 3′UTR of the AR transcript. Notably, the suppression of AR expression by miR-185 has compromised the interaction between AR and ARE and decreased the levels of the AR target gene PSA [10]. Moreover, miR-185-mediated inhibition of AR has suppressed the proliferation of prostate cancer cells and enhanced their apoptosis. Thus, the miR-185 has been suggested as a negative regulator of AR signaling and tumor suppressor miRNA in LNCaP cells [10].

MiR-381 is another down-regulated miRNA in prostate cancer. Forced over-expression of miR-381 in LNCaP cells inhibited their proliferation, migratory aptitude, and invasion. Mechanistically, miR-381 suppresses AR mRNA expression by binding to its 3′UTR [11].

MiR-let-7c is another miRNA that decreases expression and activity of AR in prostate cancer cells. It modulates AR transcription through c-Myc. MiR-let-7c-mediated inhibition of AR can reduce the proliferation of prostate cancer cells [12].

Figure 1 depicts a number of tumor suppressor miRNAs that regulate expression of AR in prostate cancer cells.

### 2.2. Indirectly Regulate AR Expression or AR Signal

On the other hand, a handful of miRNAs have been found to exert oncogenic roles in prostate cancer, through the regulation of transcription of AR expression or signaling. Following androgen deprivation therapy, hormone-sensitive prostate cancer can evolve to castration-resistant prostate cancer (CRPC). MiRNAs can contribute to this process. For instance, miR-221/-222 has been shown to be up-regulated in bone metastatic CRPC samples. In vitro studies have demonstrated that stable overexpression of miR-221 induces the androgen-independent growth of prostate cancer cells, by releasing these cells from androgen deprivation-related G1 arrest. The up-regulation of this miRNA in LNCaP has led to the reduction of expression of a subclass of androgen-responsive genes, without influencing the expression of AR or integrity of AR-androgen. MiR-221 has been found to regulate the expressions of HECTD2 and RAB1A, two genes being capable of the induction of CRPC phenotype in various prostate cancer cells. Further, the up-regulation of miR-221 has led to alterations in the expression levels of several cell cycle-related genes and the activation of EMT-related pathways. Taken together, it has been hypothesized that miR-221 has a major role in AR signaling reprogramming and the subsequent evolution of the CRPC phenotype [13].

Experiments in mice models have indicated the effect of surgical castration in the induction of an early upsurge in the serum levels of miR-125b. Moreover, bicalutamide-mediated AR blocking has resulted in the prompt release of this miRNA into the media of cultured prostate cancer cells. NCOR2, as a corepressor of AR, has been shown to be targeted by miR-125b. Thus, miR-125b has been suggested as a key regulator of AR, which alters the efficacy of anti-androgen therapies [14].

MiR-96 is another oncogenic miRNA that can target a RARγ network to control AR signaling. Down-regulation of RARγ, a member of the nuclear receptor superfamily, has been shown to significantly affect the viability of prostate cancer cells and gene signature of these cells. A gene network, comprising of numerous RARγ target genes, such as SOX15, has been found to be correlated with poor disease-free survival of prostate cancer patients [15].

MiR-541 is another oncogenic miRNA that can affect prostate cancer course through modulation of AR signaling. In fact, infiltrating CD4(+) T cells, which are associated with poor clinical outcomes in this type of cancer, can increase FGF11 levels. Up-regulation of this growth factor leads to increase levels of miR-541. The subsequent down-regulation of AR signaling regulates MMP9 levels, in favor of tumor metastasis [16]. Figure 2 shows the effects of oncogenic miRNAs in the progression of prostate cancer, through the modulation of AR signaling.

In addition to above-mentioned miRNAs, several miRNAs can directly or indirectly affect AR signaling by binding with 3′UTR, the coding region of AR, or influencing the levels of AR co-activators/co-repressors (Table 1).

**Table 1 cells-10-03198-t001:** The effects of different miRNAs on AR in prostate cancer (prostate cancer (PCa), FUS: fused in sarcoma, AR-V7: androgen receptor variant 7, BCa: breast cancer, BPH: benign prostatic hyperplasia, ANCTs: adjacent non-cancerous tissues, PEITC: phenethyl isothiocyanate, Enz: enzalutamide, CIN: cervical intraepithelial neoplasia, AI: androgen-independent, 5-hmC: 5-Hydroxymethylated cytosine, ↓: decrease in, ↑: increase in).

miRNAs	Expression of miRNAs in PCa	Target Region of AR mRNA/Effect of miRNAs on AR	Targeted Pathway	Cell Line/Samples/Animal Models	Function of miRNAs in Cancer Cells	References
miR-299-3p	↓	↓	VEGFA signaling	LNCaP-104S, MDA-PCa-2b, 22Rv-1, C4-2B, PC-3, WPE-1/TCGA PRAD publication: 330 matching tumor and 51 normal samples	↓ proliferation, EMT process and migration, growth, ↑ cell cycle arrest, apoptosis, and drug sensitivity	[9]
miR-185	↓	↓ 3′UTR	ARE, PSA	LNCaP	↓ proliferation, ↑ apoptosis	[10]
miR-381	↓	↓ 3′UTR	_	LNCaP	↓ proliferation, migration, and invasion	[11]
miR-1207-3p	↓	↓	FNDC1, FN1, AR	WPE1-NA22, MDA PCa 2b, PC-3, E006AA, E006AA-hT, LNCaP, C4-2B, RWPE-1	↓ proliferation, migration, ↑ apoptosis	[17]
miR-21	↑	↑	TGFBR2, Smad2/3	RWPE-1, MDA-PCa-2b, 22Rv1, PC-3, and LNCaP/male athymic nude mice	↓ tumor-suppressive activity of TGFβ pathway	[18]
miR-let-7c	↓	↓ suppression of AR at the level of transcription	Lin28, c-Myc	LNCaP, C4-2B/22 PCa samples/nude mice	↓ proliferation, ↓ transactivation, potential of AR	[12]
miR-133a-5p	↓	↓ 3′UTR	FUS, PSA, IGF1R, and EGFR	RWPE-1, VCaP, and LNCaP/TCGA database: 497 tumor tissue samples and 52 non-cancerous tissue samples	↓ proliferation and viability	[19]
miR-103a-2-5p/miR-30c-1-3p	↓	↓ 3′UTR AR-V7	_	VCaP	↓ cell growth and proliferation	[20]
miR-30b-3p and miR-30d-5p	↓	↓ 3′UTR	_	LNCaP, PC3, LAPC4/15 primary PCa samples, 15 adjacent normal prostate samples, and 15 metastatic CRPC samples	↓ cell growth	[21]
miR-31	↓	↓ coding region	_	RWPE-1, VCaP, LNCaP, 22Rv1, PC3, DU145, and HEK293	↓ proliferation, cell growth and colony formation, ↑ cell cycle arrest	[22]
miR-205	↓	↓ 3′UTR	_	DU145, PC3, 22Rv1, LNCaP/49 PCa, and 25 samples without PCa	↓ proliferation, colony formation and metastases, ↑ cell adhesion, overall survival	[23]
miR-124	↓	↓ 3′UTR	_	LNCaP, 22Rv1, DU145, PC-3, C4-2/male BALB/C nude mice	↓ proliferation, migration, and cell growth	[24]
miR-145	↓	↓	Ago2, PSA, TMPRSS2, KLK2	PC3, DU145, LNCaP, 22Rv1, VCaP, PNT2/49 PCa, and 25 samples without PCa	↓ proliferation, ↑ G1 arrest	[25]
miR-8080	_	↓ AR-V7 3′-UTR	IGF-1R and NKX3.1	22Rv1 and VCaP/male TRAP rats and male nude mice	Luteolin treatment: ↑ MiR-8080: ↓ proliferation, growth and oxidative stress, ↑ apoptosis, and Enz effects under castration conditions	[26]
miR-124	↓	↓ 3′UTR	p53	RWPE-1, pRNS-1-1, LNCaP, C4-2B, cds2, 22Rv1, and LAPC-4/8 matched pairs of CaP and BPH tissues/male athymic nu/nu mice	↓ cell growth, ↑ apoptosis	[27]
miR-124	↓	↓ 3′UTRs of AR-V4, -V7	EZH2 and Src	LNCaP, C4-2B, 22Rv1, and VCaP/male athymic nude mice	↓ proliferation and cell growth, ↑ apoptosis, sensitivity to Enz	[28]
miR-125b	↑	↑ indirectly by decreasing the co-repressor of AR	NCOR2	HEK293 and LNCAP/male nude mice	↑ cell growth and survival, ↓ apoptosis	[14]
miR-473p	↑	_	MEKK1	LNCap/38 pairs of tumor tissues and ANCTs	↑ cell survival, ↓ docetaxel-induced apoptosis in AR+ prostate cancer cells	[29]
miR-185	↓	↓ directly by binding 3′UTRs, ↓ indirectly by suppressing co-activator of AR	BRD8 ISO2	LNCaP, PC-3/10 pairs of tumor tissues and ANCTs	↓ proliferation and invasion	[30]
miR-449	_	↓ AR-v7	EZH2	CWR22Rv1 and VCaP/male nude mice	↓ cells growth and invasion, Enz resistance	[31]
miR-34b	↓	↓ 3′UTR	ETV1	MDA-PCa-2b, DU-145/143 PCa samples (from 3 different groups), and GEO analysis: GSE21032	↓ proliferation, ↑ apoptosis	[32]
miR-320a	↓	↓ 3′UTR	_	22Rv1, VCaP, and LNCaP/10 PCa samples/SD rats	OBP-801 treatment: ↑ miR-320a: ↓ proliferation and cell growth	[33]
miR-17	↓	↓ indirectly by suppressing co-activator of AR	PCAF	RWPE1, LNCaP, PC-3, DU145, C4–2B, and ALVA31	PEITC treatment: ↑ miR-17: ↓ cell growth	[34]
miR-141	↑	↑ AR-regulated transcriptional activity	Shp	RWPE-1, LNCaP, DU145, and C4-2B	PEITC treatment: ↓ miR-141 and AR signaling activation	[35]
miR-449a	_	↓ 3′UTR	PSA	C4-2 and LNCaP	capsaicin treatment: ↑ miR-449a: ↓ proliferation, ↑ G0/G1 cell cycle arrest	[36]
miR-331-3p	↓	↓ indirectly by regulating ERBB-2	ERBB-2, PI3K/AKT signaling pathway, PSA	LNCaP, 22RV1, DU145/tumor tissues, and ANCTs	↓ indirectly AR pathway target genes via cross-talk between ERBB-2 and AR signaling pathways	[37]
miR-371	↓	↓ 3′UTR	KLK3	LNCaP and PC3/83 PCa samples and 6 BPH as controls/male nude mice	↓ proliferation and tumor growth	[38]
miR-1207-3p	↓	↓ indirectly by regulating FNDC1	FNDC1, FN1	RWPE-1, CM, WPE1-NA22, RWPE-1, MDA PCa 2b, PC-3, E006AA, E006AA-hT, LNCaP, C4-2b	↓ proliferation, migration, ↑ apoptosis	[39]
miR-301a	↑	↓ 3′UTR	TGF-β1/Smad/MMP9 signals	CWR22Rv1, 3T3-L1/21 pairs of tumor tissues, and ANCTs/male nude mice	Recruitment of pre-adipocytes: ↑ miR-301a: ↑ invasion and metastasis	[40]
miR137	↓	↓ indirectly by regulating AR cofactor complexes	NCoA2, KDM1A, KDM2A, KDM4A, KDM5B, KDM7A and MED1	PREC, LNCaP, LNCaP:C4-2, and PC-3/TCGA database	miR137: suppressor of androgen signaling by modulating expression of transcriptional coregulators	[41]
miR-361-3p	↓	↓ 3′UTR of ARv7	_	CW22Rv1, C4-2, and LNCaP/TCGA analysis/male nude mice	↑ Enz sensitivity	[42]
miR-2909	↑	↑	TGFBR2, TGFβ signaling, PSA	PC3 and LNCaP	↑ cell growth	[43]
miR-200a	↓	↓ AR-V7 indirectly by regulating BRD4	BRD4	LNCaP and C4-2B/10 ADPC tissue and 10 CRPC tissue samples	↓ proliferation, ↑ apoptosis	[44]
miR-135b	_	↓	MUC1-C	LNCaP	↑ invasion and EMT process	[45]
miR-17-5p	↓	↓ indirectly by regulating co-activator of AR	PCAF, PSA	RWPE1, LNCaP, C4-2B, PC3, and PrEC	↓ cell growth	[46]
miR-3162-5p	↑ in PCa tissues with higher Gleason grade	↓ 3′UTR	KLK3, PSA	LNCaP, PC3	↓ proliferation, migration, and colony formation	[47]
miR-644a	↓	↓ 3′UTR (directly) and indirectly by regulating co-activators of AR	SRC-1, SRC-2, SRC-3, CCND1, CBP, and ARA24	LNCaP, LAPC4, and 22RV1/male athymic nude male mice	↓ invasion, EMT process, metastasis and Warburg effect, ↑ apoptosis	[48]
miR-221	↑	↓ indirectly by regulating co-activators of AR	HECTD2 and RAB1A	LNCaP and LNCaP-Abl, LAPC-4, PC-3, Du145, and 22Rv1	↑ AI cell growth, emt process, and metastasis	[13]
miR-29b	↑	↑ indirectly by regulating co-activators of AR	TET2, FOXA1, mTOR	LNCaP, BicR, VCaP, and 293T/male BALB/C nude mice	↑ 5-hmC-mediated tumour progression	[49]
miR-141-3p	_	↓ 3′UTR	_	LNCaP	↓ both mRNA and protein expression levels of AR	[50]
miR-96	↑	↓ indirectly by regulating co-activator of AR	RARγ, TACC1	RWPE-1, RWPE-2, PNT2, HPr1-AR, LNCaP, LAPC4, EAA006, MDAPCa2b, LNCaP-C42, 22Rv1, PC3 and DU145/36 PCa samples, and MSKCC dataset	↑ proliferation and viability	[15]
miR-185	↓	↓ indirectly by regulating co-activator of AR	SREBP signaling	LNCaP, C4-2B, RWPE-1/male athymic nude mice	↓ proliferation, clonogenicit, tumorigenicity, cell growth, migration and invasion, ↑ apoptosis	[51]
miR-342	↓	↓ indirectly by regulating co-activator of AR	SREBP signaling	LNCaP, C4-2B, RWPE-1/male athymic nude mice	↓ proliferation, clonogenicit, tumorigenicity, cell growth, migration and invasion, ↑ apoptosis	
miR-204	↓	↓ indirectly by regulating XRN1	XRN1, PSA, miR-34a	LNCaP, 22Rv1 and PC-3 and CL1/171 BPH, plus PCa samples/nude mice and rats	↓ growth and colony formation of LNCaP and 22Rv1 cells but ↑ growth and colony formation of CL1 and PC-3 cells	[52]
miR-541	↑	↓	FGF11, MMP9	LNCaP, CWR22RV1 and C4-2/20 PCa samples/male nude mice	↑ invasion and metastasis (while infiltrated T cells co-cultured with PCa cells)	[16]
miR-205	↓	↓ indirectly by regulating SQLE	SQLE	LNCaP, C4-2, PC-3, DU145, RWPE-1, HEK293T, VcaP, andLNCaP Abl	↓ cell growth and de novo cholesterol biosynthesis	[53]
miR-130a	↓	↓ indirectly by regulating coregulators of AR	CDK1, PSAP, PSMC3IP, GTF2H1	LNCaP, PC-3, Du-145 and RWPE-1/5 low Gleason grade PCa samples, 6 high Gleason grade PCa samples, 3 recurrent PCa samples, and 6 nonmalignant samples	↑ apoptosis	[54]
miR-203	↓	↓ indirectly by regulating coregulators of AR	PARK7, MNAT1, TFIIH, NCOA4, CDK1	LNCaP, PC-3, Du-145 and RWPE-1/5 low Gleason grade PCa samples, 6 high Gleason grade PCa samples, 3 recurrent PCa samples, and 6 nonmalignant samples	↑ apoptosis and cell cycle arrest	
miR-205	↓	↓ indirectly by regulating coregulators of AR	PARK7, RAN, KHDRBS1	LNCaP, PC-3, Du-145 and RWPE-1/5 low Gleason grade PCa samples, 6 high Gleason grade PCa samples, 3 recurrent PCa samples, and 6 nonmalignant samples	↑ cell cycle arrest	
miR-212	↓	↓ (AR and AR-V7) indirectly by regulating hnRNPH1	hnRNPH1, SRC-3	LNCaP, MDA-PCa-2b and C4–2B/13 African American samples, and 17 Caucasian American samples/SCID mice	↓ cell growth and ↑ sensitivity to bicalutamide	[55]
miR-34a	↓	↓ 3′UTR	Notch-1	C4-2B, CWR22rv1, LNCaP, and VCaP	↓ proliferation and self-renewal capacity	[56]
miR-190a	↓	↓ indirectly by regulating the activator of AR	YB-1	LNCaP, C4-2, PC-3, DU-145, 22Rv1/mal nude mice	↓ proliferation and cell growth	[57]

### 2.3. Effects of miRNAs on AR in Other Different Cancer Types

In addition to prostate cancer, the effects of miRNAs on AR have been investigated in other cancer types (Table 2). For instance, experiments in two AR-positive bladder cancer cell lines have shown that phenyl glucosamine can inactivate and degrade AR through the restoration of miR-449a expression. Lentivirus-mediated up-regulation of miR-449a has been shown to further suppress the proliferation of these cells via the induction of cell cycle arrest [58].

In addition, miR-9-5p has been identified as a suppressor of AR expression in breast cancer. A feedback loop has been recognized between these genes in breast cancer cells, in which androgen agonists of AR can increase expression of miR-9-5p. In fact, miR-9-5p can inhibit the proliferation of breast cancer cells in a manner independent from the estrogen receptor (ER) status of these cells. Moreover, miR-9-5p can decrease the activity of AR-downstream signals, even in the conditions that breast cancer cells are induced by AR-agonists [59].

In cervical cancer, the oncogenic miR-130a-3p has been found to target both ERα and AR. MiR-130a-3p silencing, ERα up-regulation, and AR up-regulation have suppressed proliferation and invasion of cervical cancer cells. Besides, antagomiR-130a could decrease tumor bulk in animal models. Taken together, miR-130a-3p has a possible role in the progression of cervical cancer, through the suppression of ERα and AR [60].

In hepatocellular carcinoma, miR-135b-5p has been shown to suppress AR-mediated cell proliferation, through the regulation of HIF-2α/c-Myc/P27 axis [61]. Moreover, macrophage-derived miR-92a-2-5p-containing exosomes could increase the invasiveness of liver cancer cells, through the modulation of AR/PHLPP/p-AKT/β-catenin axis [62]. Finally, miR-367-3p could increase the effectiveness of sorafenib in the suppression of liver cancer metastasis via the modulation of AR signals [63].

In glioma, the circ-ASPH/miR-599/AR/SOCS2-AS1 axis has been identified as a molecular mechanism for cancer progression. In fact, circ-ASPH could mediate this function by sponging miR-599 [64].

## 3. Regulatory Impact of lncRNAs on AR

### 3.1. Regulation of AR Expression

PCGEM1 is a lncRNA with important roles in splicing events. This lncRNA has been found to interact with the splicing factors heterogeneous nuclear ribonucleoprotein (hnRNP) A1 and U2AF65. Experiments have shown correlation between PCGEM1 and AR3, a principal and clinically important alternatively spliced form of AR in prostate cancer. Besides, androgen deprivation leads to enhanced expression of PCGEM1 and its accretion in nuclear speckles. Androgen deprivation-induced PCGEM1 has a role in regulation of the competition between two splicing factors for AR pre-mRNA [65]. Another study has identified HORAS5 as a CRPC-promoting lncRNA through assessment of patient-derived xenografts, clinical information with subsequent in vitro and in vivo confirmation studies. This lncRNA is a cytoplasmic lncRNA, which increases proliferation and viability of prostate cancer cells via sustaining AR activity even in androgen-depleted settings. Notably, HORAS5 silencing has reduced AR expression, as well as expression of oncogenic targets of AR, including KIAA0101. In clinical samples, up-regulation of HORAS5 has been associated with poor survival. Taken together, HORAS5 has been identified as targetable contributor in the induction of CRPC phenotype through maintaining oncogenic activity of AR [66]. Recent investigations identified a novel lncRNA*SAT1* as AR-interacting partner. The expression of this lncRNA is down-regulated in PCa tumor tissue compared to non-tumor tissue indicating a tumor suppressive function. LncRNA*SAT1* is up-regulated by the treatment with supraphysiological androgen level (SAL) in PCa cells and human PCa tissue ex vivo and mediates the SAL-induced cellular senescence [67]. Further, it has been shown that lncRNA*SAT1* interacts with AR on chromatin level regulating AR transactivation and AR target gene expression [68]. Another study has identified a feed-forward regulatory circuit between AR and PlncRNA-1, which enhances progression of prostate cancer [69]. In addition, PCAL7 lncRNA is another lncRNA that enhances progression of this type of cancer through promoting AR signaling [70]. Figure 3 depicts the role of a number of lncRNAs in progression of prostate cancer through modulation of AR signaling.

### 3.2. Regulation of AR Activity as AR-Interacting Partner

Several other oncogenic lncRNAs have been found to regulate AR signaling. For instance, HOTAIR increases AR-mediated transcriptional program and induces CRPC phenotypes [71]. Moreover, MALAT1 has been shown to suppress cell cycle progression in this type of cancer through regulation of AR signaling [72]. LINC00844 is another lncRNA that affects migration and invasion of prostate cancer cells through modulation of this route [73].

In brief, lncRNAs can affect AR levels through interacting with AR on chromatin level, regulation of AR transactivation and modulation of AR target gene expression. Some lncRNAs can also regulate stability of AR transcripts and preventing its ubiquitination. Table 3 shows the effects of different lncRNAs on AR in prostate cancer.

### 3.3. Effects of lnRNAs on AR in Other Different Cancer Types

The effects of lncRNAs on AR signaling have also been assessed in other types of cancers. In bladder cancer, XIST has been found to be up-regulated parallel with up-regulation of AR. Over-expression of XIST and AR has been correlated with advanced TNM stage in this cancer. XIST silencing has decreased proliferation, invasion, and migratory potential of bladder cancer through modulation of AR signaling. Mechanistically, XIST suppresses expression of miR-124 through direct interaction. Besides, miR-124 has been shown to target 3′UTR of AR [92]. Another experiment in bladder cancer has shown over-expression of LINC00460. LINC00460 levels have been correlated with poor prognosis of these patients. LINC00460 silencing decreased proliferation of 5637 and T24 bladder cancer cells. Based on the observed down-regulation of AR in bladder urothelial cancer tissues, it has been suggested that LINC00460 might exert its oncogenic roles through modulation of AR expression [93]. LINC00278, SLNCR1, SARCC and HOTAIR are other lncRNAs whose effects on AR have been investigated in different cancer types (Table 4).

## 4. Effects of circRNAs on AR

circZMIZ1 has been shown to be over-expressed in plasma samples of patients with prostate cancer compared with those having benign prostatic hyperplasia (BPH). In vitro studies have shown that circZMIZ1 silencing inhibits cell proliferation and arrests cells at G1. Functionally, circZMIZ1 enhances expression of AR and its splice variant 7 (AR-V7) [100].

On the other hand, expression of cir-ITCH has been shown to be decreased in the tissues and cell lines of prostate cancer compared to corresponding controls. Up-regulation of cir-ITCH could suppress proliferation, migratory potential, and invasiveness of human prostate cancer cells. A reciprocal inhibitory effect has been found between this circRNA and miR-17. Several molecules within Wnt/β-catenin and PI3K/AKT/mTOR cascades have been found to be influenced by cir-ITCH. This circRNA could indirectly reduce expression of AR through regulating the coactivator of this nuclear factor [101]. hsa_circ_0004870 [102] and circRNA17 [103] are two other circRNAs that reduce AR-V7 levels through U2AF65 and miR-181c-5p mediated routes, respectively, thus enhancing efficacy of enzalutamide. Table 5 shows the effects of different circRNAs on AR in prostate cancer.

## 5. Effects of AR on ncRNAs

### 5.1. AR Responsive miRNA

AR has been found to regulate the expression of several ncRNAs. For instance, activated AR has been shown to increase the expression of miR-203 and decrease the expression of SRC kinase in prostate cancer model systems. MiR-203 has a direct interaction with the 3′UTR of SRC and affects its stability following AR activation. A reduction in AR-induced miR-203 levels has been associated with an increased growth and migration potential of prostate cancer cells. The dysregulation of the AR signaling in prostate cancer cells results in the over-expression of SRC and enhancement of metastatic ability of these cells [104].

Another experiment has shown that AR represses the expression of both miR-221/-222. The derepression of their expression after androgen deprivation has enhanced proliferation of prostate cancer cells via facilitating G1/S phase transition. Although this effect might be transient, it has a possible role in the evolution of CRPC. The restoration of AR activity via AR up-regulation could subsequently down-regulate miR-221/-222 [105]. MiR-182-5p is another AR-regulated miRNA that facilitates the progression of prostate cancer by targeting the ARRDC3/ITGB4 axis [106]. Another study has shown the effects of AR on the down-regulation of miR-1 expression and subsequent suppression of TCF7. This process has been shown to participate in the evolution of resistance to androgen deprivation in this type of cancer [107].

AR has been shown to differentially affect the metastasis of prostate and breast cancers, through distinctively changing vasculogenic mimicry (VM) formation. In fact, AR can enhance miR-525-5p transcription in prostate cancer, while decreasing its transcription in breast cancer by binding to different AREs in the precursor promoter of this miRNA. NFIX and HDAC2 have been identified as co-factors of AR in prostate and breast cancer cells, respectively [108]. Figure 4 shows the impact of AR on miRNAs expressions, in the context of prostate cancer.

### 5.2. AR Responsive lncRNA and circRNA

AR has also been shown to affect expression of several lncRNAs. For instance, LINC00304 is an androgen-responsive lncRNA that induces cell cycle transition and increases the proliferation of prostate cancer cells, through the regulation of CCNA1 [109]. Moreover, the androgen-associated up-regulation of POTEF-AS1 has been shown to affect apoptosis-associated pathways, in favor of prostate cancer cells survival [110]. On the other hand, the expression of PSLNR has been shown to be decreased by androgens. This lncRNA suppresses prostate cancer progression partly through regulation of the p53-dependent axis [111]. PART1, as another androgen-regulated lncRNA, can influence the toll-like receptor pathways in this type of cancer. The expression of PART1 has been induced in prostate cancer cells treated with 5α-dihydrotestosterone, indicating that this lncRNA is directly induced by androgen [112]. Figure 5 shows the effects of AR on lncRNAs in prostate cancer.

Other studies, in the context of prostate cancer, have identified AR-regulated miRNAs and lncRNAs. Moreover, a number of circRNAs, such as circRNA-51217, circRNA-ARC1, and circZMIZ1, have been found to be influenced by AR signaling (Table 6). A recent study has identified more than 3000 androgen-responsive circRNAs, using a microarray technique. Notably, the expression of more than 1000 of these circRNAs has been consistent with the expression of their parent genes, suggesting that AR may modulate their expression at the transcriptional level [113].

In brief, the effect of AR on the expression of ncRNAs is mainly associated with its role as a transcription factor.

Effects of AR on expression of ncRNAs are also implicated in the pathoetiology of bladder, breast, liver, renal, and gastric cancers (Table 7). Yet, these effects are largely context-dependent. For instance, AR could reduce the expression of miR-21 in breast cancer [155], while inducing its expression in hepatocellular carcinoma [156].

## 6. Discussion

AR has an essential role in the pathogenesis of human cancers, particularly prostate cancer. Since it is required for the development of prostate cancer, androgen deprivation therapy is regarded as a treatment for this type of cancer. Thus, the identification of the regulatory mechanisms of AR signaling is important in the design of treatment options. The importance of this process is further highlighted by the fact that castration resistance might occur during the course of treatment, as a result of expression of constitutively active AR splice variants [172], whose expressions can be modulated by ncRNAs.

Integrative transcriptomic analyses of diverse cancer cell lines and tissues have resulted in the identification of several AR-interacting ncRNAs. In the current study, we have listed ncRNAs that affect expression of AR, as well as those being affected by AR. Notably, mutual interactions have been identified between AR and some of these non-coding transcripts. For instance, the expression of the lncRNA ARLNC1 has been shown to be enhanced by the AR protein. Conversely, ARLNC1 can increase the stability of the AR mRNA through RNA-RNA interaction [74].

AR-targeting miRNAs have been suggested as potent tumor suppressors in prostate cancer. However, a number of other miRNAs have also been found to induce CRPC, by changing the activity of AR signaling. Moreover, AR signaling can affect the expression of miRNAs through different mechanisms, including feedback loops.

LncRNAs and circRNAs that regulate AR signaling have been found to interact with miRNAs. MALAT1/miR-320, CCAT1/miR-28-5P, PlncRNA-1/miR-34c, PlncRNA-1/miR-297, XIST/miR-24, SARCC/miR-143-3p, circ-ITCH/miR-17, and circRNA17/miR-181c-5p are examples of the cooperation between lncRNAs/circRNAs and miRNAs in the regulation of AR signaling. Similarly, AR-regulated lncRNAs and circRNAs have been shown to influence the expression or bioavailability of miRNAs, adding novel layers of complexity in this interaction network.

Taken together, the data presented above indicates the complexity of the transcriptional regulation of miRNAs by AR and the effects of AR on them. Moreover, the interactions between ncRNAs and AR signaling can be context-dependent.

## Figures and Tables

**Figure 1 cells-10-03198-f001:**
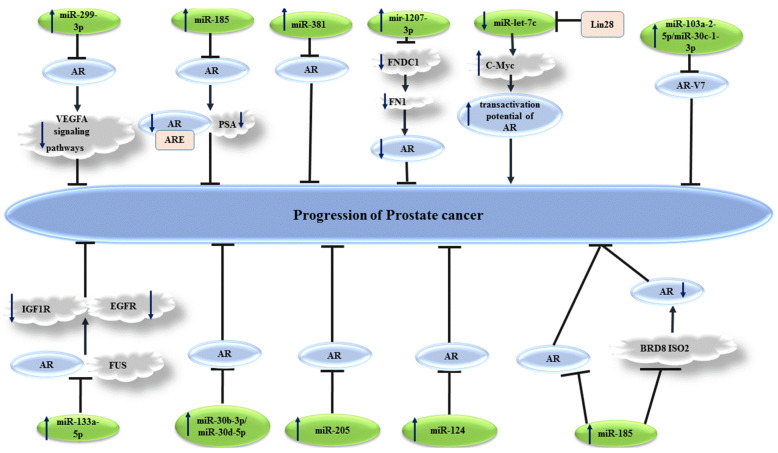
Several miRNAs have been shown to affect levels of androgen receptor (AR), thus influencing the progression of prostate cancer. Detailed information about these miRNAs is presented in Table 1. ( 
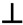
 reduction or inhibition of, 
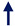
 increased levels of, 
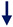
 decreased levels of).

**Figure 2 cells-10-03198-f002:**
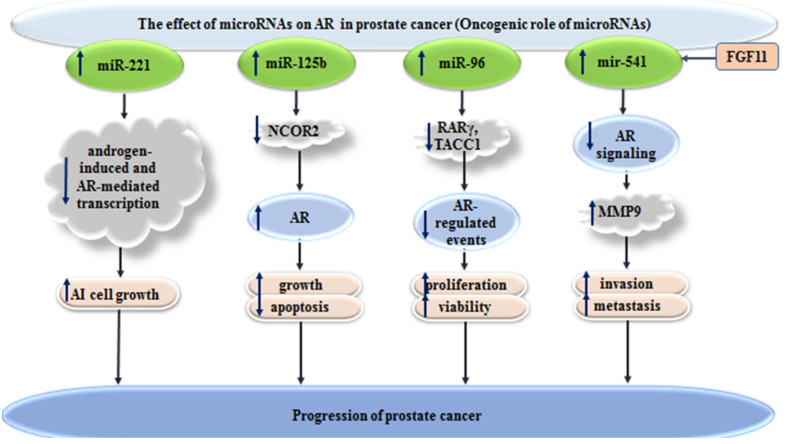
Effects of oncogenic miRNAs in progression of prostate cancer, through the modulation of AR signaling ( 
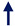
 increased levels of, 
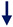
 decreased levels of).

**Figure 3 cells-10-03198-f003:**
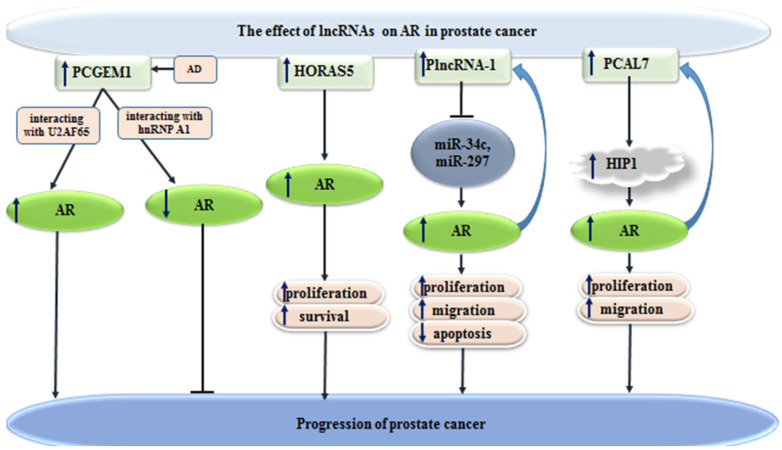
Effects of lncRNAs on AR in prostate cancer ( 
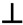
 reduction or inhibition of, 
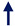
 increased levels of, 
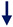
 decreased levels of).

**Figure 4 cells-10-03198-f004:**
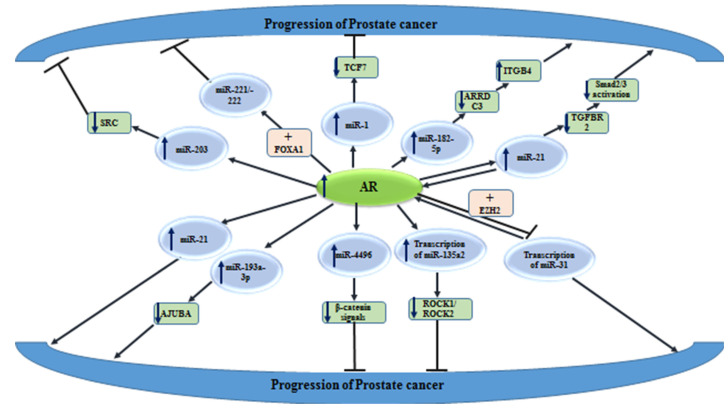
Effects of AR on miRNAs expressions, in the context of prostate cancer. Detailed information about these miRNAs is shown in Table 6 ( 
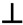
 reduction or inhibition of, 
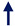
 increased levels of, 
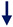
 decreased levels of).

**Figure 5 cells-10-03198-f005:**
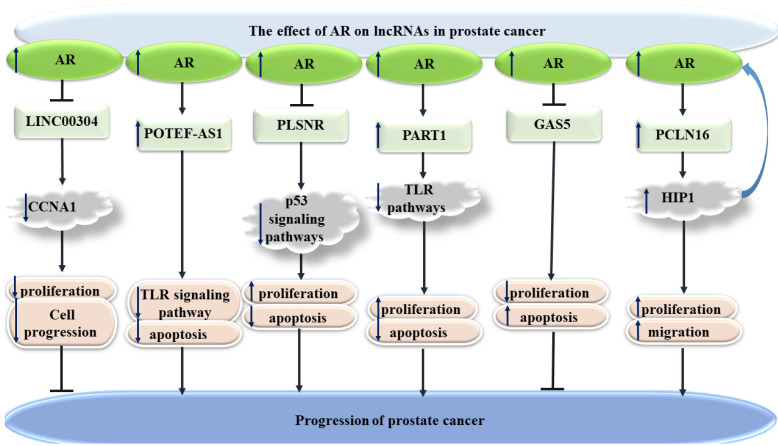
Effects of AR on lncRNAs in prostate cancer ( 
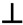
 reduction or inhibition of, 
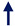
 increased levels of, 
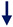
 decreased levels of).

**Table 2 cells-10-03198-t002:** Effects of miRNAs on AR in different cancer types (↓: decrease in, ↑: increase in).

Cancer Types	MiRNAs	Expression of miRNAs in Different Cancer Types	Target Region of AR mRNA/How microRNAs Affect AR	Molecular Mechanisms	Cell Line/Samples/Animal Models	Function of miRNAs in Cancer Cells	References
Bladder cancer	miR-449a	_	↓	_	UMUC3 and TCCSUP	ABDHFA treatment: ↑ miR-449a: ↓ proliferation, viability, ↑ cell cycle arrest	[58]
Breast cancer	miR-9-5p	↓	↓ 3′-UTR	_	MDA-MB-453, MCF-7, T-47D/11 pairs of tumor tissues and ANCTs	↓ proliferation and cell growth	[59]
Cervical cancer	miR-130a-3p	↑	↓ 3′UTR	_	20 CIN I, 20 CIN II, 30 CIN III tissue and 20 healthy tissue samples	↑ proliferation and invasion	[60]
Hepatocellular carcinoma	miR-135b-5p	↓	↓ 3′-UTR	HIF-2α, c-Myc, p27	SK-hep1, HepG2, SNK, Huh7 and HA22T	↓ proliferation, colony formation	[61]
	miR-92a-2-5p	↑	↓ 3′UTR	PHLPP/p-AKT/β-catenin signaling	SK-HEP-1, Hep G2, HEK 293 T, THP-1, Hepa 1-6, HA22T/male nude mice	↑ invasion	[62]
	miR-367-3p	↓	↑ indirectly by regulating MDM2	MDM2/FKBP5/PHLPP/(pAKT and pERK) signals	SKhep1 and HA22T/126 HCC samples	↑ Sorafenib chemotherapy efficacy, ↓ invasion and metastasis	[63]
Glioma	miR-599	↓	↓ 3′UTR	circ-ASPH, SOCS2-AS1	U251, U87MG, LN229	↓ proliferation, migration and invasion	[64]

**Table 3 cells-10-03198-t003:** Effects of different lncRNAs on AR in prostate cancer (ANCT: PCa: prostate cancer, DHT: dihydrotestosterone, CRPC: castration-resistant prostate cancer, ADPC: androgen-dependent prostate cancer, AD: androgen deprivation, DIM: 3,3′-Diindolylmethane, SAM: Synergistic activation mediator, LBD: ligand-binding domain, ↓: decrease in, ↑: increase in).

LncRNAs	Expression of lncRNAs in PCa	Target Region of AR mRNA/How lncRNAs Affect AR	Molecular Mechanisms	Cell Lines/Samples/Animal Models	Function of lncRNAs in Cancer Cells	References
ARLNC1	↑	stabilizing AR transcript	_	VCaP and LNCaP/11 benign prostate samples, 85 localized prostate cancer samples, and 37 from metastatic PCa samples/athymic nude mice	↑ Proliferation and cell growth, ↓ apoptosis	[74]
PRNCR1 and PCGEM1	↑	interact with, and increase its ligand-independent activation	DOT1L	LNCaP, RWPE, WPE, CWR22Rv1/BPH and PCa tissues male athymic Nu/Nu mice	↑ Proliferation and cell growth	[75]
HOTAIR	↑	↑ By preventing AR ubiquitination and blocking its interaction with MDM2	_	LNCaP, C4-2B/GEO analysis: GSE35988 and GSE21034	↑ cell growth and invasion	[71]
MALAT1	↑	↑ indirectly by inhibiting miR-320	miR-320	DU145, 22Rv1, PC3, LNCaP/BALB/cA-nu mice	DHT treatment: ↑ proliferation and cell cycle progression	[72]
LINC00844	↓	modulated AR binding to chromatin	NDRG1	LNCaP, VCaP, and 22Rv1/GEO database: GSE109336	↓ migration and invasion	[73]
LINC00675	↑	directly modulate AR interaction with MDM2, inhibited AR’s ubiquitination, ↑ indirectly by regulating the co-activator of AR	MDM2, GATA2	LNCaP-SF, LNCaP-JP, LNCaP, LNCaP-C4-2b, 293T/male BALB/c nude mice	↑ tumor formation, tumor growth and Enz resistance	[76]
CCAT1	↑	↑ by binding to P68	DDX5 (P68), mir-28-5P	PC3, Du145, and LNCaP/8 ADPC tissues and 4 CRPC samples/BALB/C nude mice	↑ proliferation, colony formation, and cell cycle progression, ↓ apoptosis	[77]
PCGEM1	↑ in AD	↑ AR3 by interacting with U2AF65, ↓ AR3 by interacting with hnRNP A1	U2AF65, hnRNP A1	LNCaP, CWR22Rv1, LNCaP95, HECK293T/male SCID mice	↑ castration resistance	[65]
SOCS2-AS1	↑ Castration-resistant Prostate Cancer Cells	↑ by regulating cofactor recruitment for epigenetic controls	TNFSF10	LNCaP, VCaP, LTAD	↑ castration-resistant and cell growth, ↓ apoptosis	[78]
HORAS5	↑	↑ post-transcriptional maintenance of AR mRNA stability	_	LNCaP and C4-2 male, immunocompromised NOD/SCID mice	↑ proliferation and survival	[66]
PCLN16	↑	↑ indirectly by regulating HIP1	HIP1	NCaP and VCaP/tumor tissues and ANCTs	↑ proliferation, migration and cell growth	[79]
HOTAIR	↑ in PCa cells after co-culture with HMC-1 cells	↓ at the transcriptional level	PRC2, MMP9	LNCaP, CWR22Rv1, C4-2, C4-2B and HMC-1/male nude mice	recruitment of mast cells: ↑ invasion and stem/progenitor cell population	[80]
PlncRNA-1	↑	↑ by sponging AR-targeting microRNAs	miR-34c and miR-297	RWPE-1, 22RV1, LNCaP, PC3 and DU145/16 PCa tissue samples, 35 biopsy-negative and 37 biopsy-positive blood samples/male nude mice	↑ proliferation, migration and viability, ↓ apoptosis	[69]
PCAL7	↑	↑ indirectly by regulating HIP1	HIP1	104 tumor tissues and ANCTs	↑ proliferation, migration	[70]
Malat1	↑	↑ AR-v7 indirectly by interacting with SF2 to splice the AR transcript	SF2	VCaP and EnzR-PCa C4-2/ 10 CRPC samples before (Pre-Enz) and after (Post-Enz) Enz treatment/nude mice	↑ Enz resistance	[81]
PlncRNA-1	↑	↑	NKX3-1	LNCaP, LNCaP-AI, PC-3, C4-2, RWPE-1 and PWR-1E/16 pairs of PCa tissues and ANCTs, 14 pairs of PCa tissues and BPH tissues	↑ proliferation and viability, ↓ apoptosis	[82]
LBCS	↓	↓ 5′ UTR	hnRNPK	LNCaP, LNCaP-Bic, and LNCaP-AI/130 PCa tissues and 32 BPH tissues plus 70 PCa tissues and 10 BPH	↓ castration resistance	[83]
PCGEM1	↑	upregulation of PCGEM1 by SAM: ↑ AR3	p54/nrb	LNCaP and CWR22Rv1/male SCID mice	↑ tumor growth and castration resistance, ↓ apoptosis DIM: ↓ PCGEM1-mediated castration resistance	[84]
PCGEM1	↑	facilitating AR binding to some promoters	c-Myc	LNCaP,	↑ glucose uptake and glycolysis, ell-cycle progression, proliferation, and survival	[85]
LOC283070	↑	↑ indirectly by inhibiting PHB2	PHB2	LNCaP and LNCaP-AI	↑ proliferation and migration	[86]
lnc-OPHN1-5	_	↓ 3′UTR	hnRNPA1	C4-2R, C4-2BR, C4-2B/75 PCa samples/male NOD CRISPR Prkdc Il2r Gamma triple-immunodeficient mice	↑ Enz sensitivity	[87]
GAS5	↓	↓ directly by interacting with LBD of AR	_	C4-2, DU145, 293T/GSE6919	↓ proliferation, ↑ apoptosis	[88]
GHSROS	↑	↓	PPP2R2C	PC3, LNCaP, DU145, DUCaP	↑ proliferation, growth, migration, survival, and resistance to the cytotoxic drug docetaxel	[89]
PCA3	↑	PCA3 knock down→ ↑ regulation of AR cofactors	_	LNCaP	modulating the expression of EMT markers and AR cofactors ∆ PCA3: ↓ cell viability	[90]
PRNCR1	↑	↑	_	LNCaP and C4-2	↑ proliferation and invasion, ↓ apoptosis	[91]

**Table 4 cells-10-03198-t004:** Effects of lncRNAs on AR in different cancer types (ND: nutrient deprivation, ccRCC: clear cell renal cell carcinoma, ↓: decrease in, ↑: increase in).

Cancer Types	LncRNAs	Expression of LncRNAs in Different Cancer Types	Target Region of AR mRNA/How lncRNAs Affect AR	Molecular Mechanisms	Cell Lines/Samples/Animal Models	Function of lncRNAs in Cancer Cells	References
Bladder cancer	XIST	↑	↑ by sponging AR-targeting microRNA	miR-124	TCC-SUP, EJ, SW780 and UM-UC-3, SV-HUC-1 67 pairs of tumor tissues and ANCTs	↑ proliferation, migration and invasion	[92]
	LINC00460	↑	↓	_	5637, T24, J82, TCCSUP, UM-UC-3 and SV-HUC-1/TCGA database	↑ proliferation and migration	[93]
Esophageal squamous cell carcinoma	LINC00278	↓	indirectly inhibited interaction between YY1 and AR	YY1, eEF2K, YY1BM	DMEM, RPMI1640, FBS, Eca-109, TE-1, and KYSE-30/281 pairs of ESCC tissues and ANCTs,	ND treatment: ↑ LINC00278: ↓ survival, ↑apoptosis	[94]
Melanoma	SLNCR1	↑	SLNCR1 binds to AR-binding motifs 1 and 2	_	A375, HEK293T, WM1976,	↓ binding SLNCR1 to AR: ↓ SLNCR1-mediated invasion	[95]
	SLNCR1	↑	↑ AR binding to the MMP9 promoter	Brn3a, MMP9	A375, HEK293T, CY and WM	↑ invasion	[96]
Renal cell carcinoma	SARCC	↓	destabilizing AR protein	miR-143-3p, AKT, MMP-13, K-RAS and P-ERK	SW839, OSRC-2, A498, 769-P, 786-O, Caki-1, Caki-2, HK2/66 ccRCC tissues and 8 metastatic ccRCC tissues/male athymic nude mice	↓ proliferation, invasion, migration and resistance to Sunitinib	[97]
	SARCC	Differentially expressed by hypoxia in a VHL-dependent manner	↓ binding and destablizing AR protein	HIF-2α, C-MYC signals	SW839, OSRC-2, A498, 769-P, and 786-O, Caki-1, Caki-2, HK-2 and 293T/16 ccRCC samples/male athymic nude mice	Differentially modulates proliferation under hypoxia	[98]
	HOTAIR	↑	↑	GLI2	HK-2, 786-O, ACHN, 769-P, SW839, OSRC-2, HUVEC/male nude mice	↑ angiogenic phenotype and stemness	[99]

**Table 5 cells-10-03198-t005:** The effects of different circRNAs on AR in prostate cancer (HCs: healthy controls, Enz: enzalutamide, ↓: decrease in, ↑: increase in).

circRNAs	Expression of circRNAs in PCa	Target Region of AR mRNA/How circRNAs Affect AR	Regulated Pathway	Cell Lines/Samples/Animal Models	Function of circRNAs in Cancer Cells	References
circZMIZ1	↑	↑ AR and AR-V7	_	DU145, C4-2, LNCaP, 22RV1, RWPE-1, 14 PCa samples, and 14 HCs	↑ proliferation, ↓ G1 arrest	[100]
circ-ITCH	↓	↓ indirectly by regulating the coactivator of AR	miR-17, Wnt/β-Catenin, and PI3K/AKT/ mTOR Signaling Pathways	RWPE-1, LNCaP, PC-3/10 pairs of tumor tissues and ANCTs	↓ migration and invasion	[101]
hsa_circ_0004870	↓	↓ AR-V7 indirectly through U2AF65	RBM39, U2AF65	LNCaP, BPH1, 22Rv1	↓ Enz resistance	[102]
circRNA17	↓	↓ AR-v7 indirectly by regulating miR-181c-5p	miR-181c-5p	C4–2, CWR22Rv1, and 293T/13 BPH samples, and 14 PCa samples/male nude mice	↓ Enz resistance and invasion	[103]

**Table 6 cells-10-03198-t006:** Effects of AR on different ncRNAs in prostate cancer. (PCa: prostate cancer, CRPC: castration-resistant prostate cancer, HCs: healthy controls, NE: neuroendocrine, BPH: benign prostatic hyperplasia, CRPC: castration-resistant prostate cancer, DOX: doxorubicin, NED: neuroendocrine differentiation, NE: neuroendocrine, Enz: enzalutamide, PRAD: prostate adenocarcinoma, R-2HG: R-2-hydroxyglutarate, ↓: decrease in, ↑: increase in).

ncRNAs	Regulation by AR	Molecular Mechanisms	Cell Line/Samples/Animal Models	Function of ncRNAs in Cancer Cells	References
miR-203	↑	SRC	_	↓ migration, growth, and metastasis	[104]
miR-221/-222	↓	FOXA1	LNCaP and C4-2B/LuCaP 35 and LuCaP 35CR xenografts	↑ proliferation and development of CRPC	[105]
miR-182-5p	↑	ARRDC3, ITGB4	RWPE-1, 22RV1, LNCaP, DU145/65, pairs of tumor tissues and ANCTs, and 18 pairs of tumor tissues and ANCTs/male nude mice	↑ proliferation, invasion, migration and growth, ↓ apoptosis	[106]
miR-1	↑	TCF7	PC3, LNCaP/111 PCa samples/nude mice	↓ proliferation	[107]
miR-525-5p	↑	SLPI, NFIX	_	↓ PCa metastasis	[108]
miR-21	↑	TGFBR2, Smad2/3	RWPE-1, MDA-PCa-2b, 22Rv1, PC-3, and LNCaP/male athymic nude mice	↓ tumor-suppressive activity of TGFβ pathway	[18]
miR-21	↑	_	LNCaP, LAPC-4, C4-2, CWR22Rv1/10 PCa samples/male athymic Nu/Nu mice	↑ androgen-dependent and -independent proliferation, tumor growth, and castration resistance	[114]
miR-193a-3p	↑	AJUBA	LNCaP, C4-2B	↑ migration and metastasis	[115]
miR-4496	↑	β-catenin signals	C4-2 and PC3	↓ invasion	[116]
miR-135a	↑	ROCK1 and ROCK2	LNCaP, PC-3/56 pairs of tumor tissues, and ANCTs/chick embryos and adult male mice	↓ invasion	[117]
miR-31	↓	EZH2	RWPE-1, VCaP, LNCaP, 22Rv1, PC3, DU145, and HEK293	↓ proliferation, cell growth and colony formation, ↑ cell cycle arrest	[22]
miR-421	↓	NRAS, PRAME, CUL4B, and PFKFB2	LNCaP, 22Rv1, PC-3 and DU 145/microarray data: GSE21036, GSE45604, GSE38241, and 13 PCa samples 11 samples without PCa	↓ viability, glycolysis and migration, ↑ cell cycle arrest	[118]
miR-1	↑	SRC	LNCaP, DU145RasV12G37, DU145/RasB1/28 HCs, 98 primary tumor, and 13 distant metastasis samples/male nude mice	↓ proliferation, invasion, and metastasis	[119]
miR-32	↓	NSE	RWPE1, LNCaP, and CWR22Rv1/male nude mice	enzalutamide treatment (mast cells) → suppression of AR: ↑ miRNA32: ↑ NE differentiation	[120]
miR-21 promoter	↑	PDCD4	LNCaP and HEK 293, LAPC4/male athymic nu/nu mice	↑ androgen-dependent and -independent growth and castration resistance, ↓ apoptosis	[121]
miR-22	↓	LAMC1	LNCaP, PC3, DU145, VCaP, CWR22RV1, DUCaP, BPH-1, PC3-AR, LAPC-4, RWPE-1, and EP156T/ 41 pairs of tumor tissues and ANCTs, TCGA analysis: 52 pairs of tumor tissues and ANCTs	↓ migration	[122]
miR-29a	↓	MCL1	LNCaP, PC3, DU145, VCaP, CWR22RV1, DUCaP, BPH-1, PC3-AR, LAPC-4, RWPE-1, and EP156T/ 41 pairs of tumor tissues and ANCTs, TCGA analysis: 52 pairs of tumor tissues and ANCTs	↓ migration and viability, ↑ apoptosis	
miR-99a/let7c/125b-2 cluster	↓	IGF1R	LNCaP, C4-2, and PC3	↓ proliferation	[123]
miR-2909	↑	TGFBR2, TGFβ signaling, PSA	PC3 and LNCaP	↑ cell growth	[43]
miR-32	↑	BTG2	LNCaP/ 5 BPH and 28 PCs, plus 7 BPH and 14 CRPCs	↑ cell growth	[124]
miR-148a	↑	PIK3IP1	LNCaP/ 5 BPH and 28 PCs, plus 7 BPH and 14 CRPCs	↑ cell growth and the number of cells in the S phase	
miR-194	↓	FOXA1, ERK Signaling	LNCaP, PC3, and 22RV1	↑ EMT process, migration, invasion and epithelial-neuroendocrine transdifferentiation	[125]
miR-27a (miR-23a27a24-2cluster)	↑	PHB	HeLa, Cos-1, LNCaP, DuCaP, VCaP, C42, DU145, PC3, and PC3wtAR	↑ cell growth	[126]
miR-200b	↑	_	PC3/male athymic mice	↓ proliferation, invasion, cell growth, EMT process and metastasis	[127]
miR-19a	↑	SUZ12, RAB13, SC4MOL, PSAP, and ABCA1	LNCaP	↑ cell viability	[128]
miR-27a	↑	ABCA1 and PDS5B	LNCaP	↑ cell viability	
miR-133b	↑	CDC2L5, PTPRK, RB1CC1, and CPNE3	LNCaP	↑ cell viability	
miR-22	↑ and ↓ during two different mechanisms	IL-6, AR c-MYC, miR-22, PHF8, KDM3A (↑) and AR c-MYC, miR-22, PHF8, KDM3A (↓)	LNCaP-Abl, LNCaP-IL-6, LNCaP/male mice	↑ sensitivity LNCaP-Abl cells to the enzalutamide treatment, ↓ proliferation	[129]
miR-17-92a	↑	ATG7	NCaP, 22Rv1, DU145, and PC-3	↓ autophagy induced by celastrol treatment	[130]
miR-204	↓	XRN1, PSA, miR-34a	LNCaP, 22Rv1, PC-3, and CL1/171, BPH, plus PCa samples/nude mice and rats	↓ growth and colony formation of LNCaP and 22Rv1 cells but ↑ growth and colony formation of CL1 and PC-3 cells	[52]
miR-34	miR-34a ↑ after DOX, but did not change with si-AR, miR-34c↑ after DOX, but to a small extent changed with si-AR	p53, SPAK, MDC1, and CaMKII	LNCaP, C4-2b, PC3, and DU145	↑ caspase activity and apoptosis	[131]
miR-135a	↑	MMP11, RBAK, PI3K/AKT pathway	LNCaP, 22RV1, DU145, PC-3, and WPMY-1	↓ proliferation and migration, ↑ cell cycle arrest, and apoptosis	[132]
the miR-200 family, miR-17-92 cluster, and miR-99a/let-7c/miR-125b-2 family	↓	HOXC6 and NKX2-2	RWPE-1 and LNCAP	↓ metastasis and EMT process	[133]
miR-101	↑	_	LNCaP, 22Rv1, DU145, and PC-3	↓ celastrol-induced autophagy	[134]
miR-27a	↓	MAP2K4, PI3K signalingpathways	TCGA database: GSE45604 andGSE21036	↓ proliferation and migration, ↑ apoptosis	[135]
miR-190a	↓	YB-1	LNCaP, C4-2, PC-3, DU-145, 22Rv1/mal nude mice	↓ proliferation and cell growth	[57]
ARLNC1	↑	_	VCaP and LNCaP/11 benign prostate samples, 85 localized prostate cancer samples, and 37 from metastatic PCa samples/athymic nude mice	↑ Proliferation and cell growth, ↓ apoptosis	[74]
PRCAT38	↑	TMPRSS2, FOXA1	LNCaP, DU145, and VCaP/20 samples (HCs and PCa)	↑ cell growth and migration	[136]
H19	↓	_	LNCaP	Enzalutamide treatment: ↑ H19	[137]
GAS5	↓	_	PC3 and 22Rv1	dexamethasone treatment in AR- PCa cell line PC3: ↑ GAS5: ↓ proliferation, ↑ G0/G1 cell arrest	[138]
p21	↓ AR binding to the ARE5, ↑ AR binding to the AGRE	EZH2, STAT3	C4-2, CWR22RV1, NE1.8, NCI-H660, and DU145	Enz treatment: ↓ AR binding to the ARE5 region of p21: ↑ p21: ↑ NED, NE-like structure	[139]
LINC00304	↓	CCNA1	LNCaP, 22RV1, DU145, PC-3, and WPMY-1/GEO database: GSE38241: 18 PCa samples and 21 HCs	↑ proliferation and cell cycle progression	[109]
POTEF-AS1	↑	TLR signaling pathway	LNCaP, VCaP, LTAD, and VCaP-LTAD	↑ cell growth, ↓ apoptosis	[110]
PLSNR	↓	p53 signaling pathway	LNCaP, 22RV1, DU145, PC-3, and WPMY-1/GEO database: GSE55909, 3 pairs of tumor tissues and ANCTs, 13 tumor tissues and ANCTs, plus 20 pairs of PCa sample	↑ proliferation and cell-cycle progression, ↓ apoptosis	[111]
PART1	↑	TLR pathways	LNCaP and PC3/30 pairs of tumor tissues and ANCTs	↑ proliferation, ↓ apoptosis	[112]
GAS5	↓	_	LNCaP, 22RV1, DU145, PC3, WPMY-1 14 tumor tissues, and 11 normal tissues	↑ proliferation, ↓ apoptosis	[140]
PCLN16	↑	HIP1	NCaP and VCaP/tumor tissues and ANCTs	↑ proliferation, migration, and cell growth	[79]
PlncRNA-1	↑	miR-34c and miR-297	RWPE-1, 22RV1, LNCaP, PC3 and DU145/16 PCa tissue samples, 35 biopsy-negative and 37 biopsy-positive blood samples/male nude mice	↑ proliferation, migration and viability, ↓ apoptosis	[69]
PCAL7	↑	HIP1	104 tumor tissues and ANCTs	↑ proliferation, migration	[70]
PCGEM1	↑	_	131 primary PCa, 19 metastasized PCa, and 29 normal prostatic tissue samples/intact mice	↑ PCGEM1 in primary PCa Androgen receptor regulated PCGEM1 in vivo.	[141]
DRAIC	↓	_	VCap, PC3M-luc	↓ transformation of cuboidal epithelial cells to fibroblast-like morphology, migration, and invasion	[142]
PCAT29	↓	_	VCap, PC3M-luc	↓ migration and metastasis	
a subset of TPCATs, most notably EPCART	↑	ERG	LNCaP, VCaP, and DuCaP/87 prostatectomy-treated samples	↑ proliferation, migration	[143]
PCAT29	↓	_	VCaP, LNCaP, and DU145/GEO database: GSE58397/male nude athymic BALB/c nu/nu mice	↓ proliferation, migration	[144]
Malat1	↓	_	VCaP and EnzR-PCa C4-2/ 10 CRPC samples, before (Pre-Enz) and after (Post-Enz) Enz treatment/nude mice	Enz treatment: ↑ Malat1	[81]
PlncRNA-1	↑	NKX3-1	LNCaP, LNCaP-AI, PC-3, C4-2, RWPE-1 and PWR-1E/16 pairs of PCa tissues and ANCTs, 14 pairs of PCa tissues and BPH tissues	↑ proliferation and viability, ↓ apoptosis	[82]
CTBP1-AS	↑	PSF, CTBP1	VCaP, LNCaP, DU145, RWPE and PrEC/105 PCa samples	↑ castration-resistant tumour growth	[145]
RP11-783K16.13, RP11-228B15.4, and CTD-2228K2.7	↑	_	GEPIA dataset	Higher expression of the lncRNAs were significantly correlated with shorter DFS time in PRAD.	[146]
PCAT1	↑ rs7463708 increases binding ONECUT2 and AR to the PCAT1 promoter	ONECUT2, LSD1, GNMT, and DHCR24	LNCaP, LNCaP/shPCGEM1/TCGA dataset	↑ proliferation and tumor growth	[147]
FAM83H-AS1	↑	miR-15a, CCNE2	LNCaP, LNCaP-AI, and DU145/GEPIA data sets: GSE513217 and GSE55062, plus 20PCa and 8 normal samples	↑ proliferation, migration, and cell cycle progression	[148]
DANCR	↓	TIMP2/3	CWR22Rv1, PC-3, and C4-2B/nude mice	↑ migration and invasion	[149]
GAS5	↓	_	LNCaP/GSE22606	↓ proliferation, ↑ apoptosis	[88]
PCAT18	↑	PES	LNCaP, C4-2, BPH/131 PCa, and 29 normal samples/NOD/SCID mice	↑ proliferation, migration, and invasion	[150]
RP1-4514.2, LINC01138, and SUZ12P1	↓	_	22RV1, DU145, PC-3 and LNCaP, WMPY-1/3 tumor tissues and 11 ANCTs, plus 14 tumor tissues, and 11 ANCTs and TCGA database	_	[151]
KLKP1	↑	_	22RV1, DU145, PC-3 and LNCaP, WMPY-1/3 tumor tissues, 11 ANCTs plus 14 tumor tissues, and 11 ANCTs and TCGA database	_	
TMPO-AS1	↓	_	LNCaP, DU145, 22Rv1, PC-3, and WPMY-1/54 pairs of PCa samples and TCGA data	↑ proliferation and migration, ↓ apoptosis	[152]
circRNA-51217	↓	R-2HG, miRNA-646, TGFβ1/p-Smad2/3 signaling, ADAR2	C4-2, PC3, DU145, LNCaP, and HEK293T/TCGA database	IDH1 mutation And R-2HG: ↑ circRNA-51217: ↑ invasion	[153]
circRNA-ARC1	↓	miR-125b-2-3p/miR-4736/PPARγ/MMP-9 signals	CWR22Rv1 and C4-2	Enz treatment: ↑ invasion	[154]
circZMIZ1	↑	_	DU145, C4-2, LNCaP, 22RV1, RWPE-1, 14 PCa samples, and 14 HCs	↑ proliferation, ↓ G1 arrest	[100]

**Table 7 cells-10-03198-t007:** Effects of AR on ncRNAs in different cancer types (Enz: enzalutamide, VM: vasculogenic mimicry, DHEA: dehydroepiandrosterone, RBM: ccRCC bone metastases, ↓: decrease in, ↑: increase in).

Cancer Types	ncRNAs	Regulation by AR	Molecular Mechanisms	Cell Line/Samples/Animal Models	Function of ncRNAs in Cancer Cells	References
Bladder cancer	miR-525-5p	↓	SLPI, HDAC2	_	↑ metastasis	[108]
	circFNTA	↑	ADAR2, miR-370-3p, FNTA pathway, KRAS signaling	SVHUC, T24, J82, 5637, and UMUC3/male athymic BALB/c nude mice	↑ invasion, metastases, and cisplatin chemo-resistance	[157]
	circRNA-ARC1	↑	miR-125b-2-3p/miR-4736/PPARγ/MMP-9 signals	T24 and UMUC3	Enz treatment: ↓ invasion	[154]
Breast cancer	let-7a	↑	CMYC and KRAS	MCF-7, MDA-MB-453, and MDA-MB-231/24 breast cancer samples	↓ proliferation, cell growth	[158]
	miR-21	↓	_	MCF-7, ZR-75-1, MDA-MB-231, SKBR3, and LNCap	↑ proliferation (miboleron: ↓ miR-21: ↓ proliferation)	[155]
Triple-negative breast cancer	ARNILA	↓	miR-204, Sox4	MDA-MB-231 and Hs578T, MDA-MB-436/88 TNBC samples/female BALB/c nude mice	↑ migration, invasion, and EMT process	[159]
Early Hepatocarcinogenesis	miR-216a	↑	TSLC1	HepG2/48tumor tissues and 13 non-tumor tissues/male athymic nude mice	↑ proliferation and migration	[160]
Hepatocellular carcinoma	miR-21	↑	PDCD4, ERβ	HepG2, HBEC2-KT/male C57BL/6 mice	DHEA: ↑ miR-21: ↑ proliferation	[156]
	miR-146a-5p	↓	BRCA1 and BCL2	SK-HEP-1 and HepG2/TCGA database analysis/male nude mice	Enz plus Olaparib treatment: ↑ miR-146a-5p: ↓ proliferation, cell growth, and viability	[161]
	circRNA7	↓	miR-7-5p, VE-Cadherin, Notch4	SKhep1, HA22T	↑ formation of VM	[162]
	circ-LNPEP	↓	miR-532-3p, RAB9A	HA22T, SK-HEP-1, and 293/male nude mice	↑ invasion and metastasis	[163]
	circARSP91	↓	ADAR1	MHCC-97h, LM3 and LO2, HEK-293T/83 pairs of tumor tissues, and ANCTs/nude mice	↓ tumor growth	[164]
Melanoma	miR-539-3p	↑	MITF-AXL signals, USP13	A375 and WM115 and C32/102 melanoma tissue samples/male nude mice	↑ invasion and metastasis	[165]
	SLNCR	SLNCR and AR cooperatively regulate several growth-regulatory genes.	p21, EGR1, MMP9	WM1976 or A375, SK-MEL-28, WM858	↑ proliferation and invasion	[166]
	SLNCR1	↑	Brn3a, MMP9	A375, HEK293T, CY and WM	↑ invasion	[96]
Cholangiocarcinoma	ZEB1-AS1	↑	miR-133b, HOXB8	HIBEC, QBC939, CCLP-1, RBE, TFK-1/54 pairs of tumor tissues, and ANCTs/female BALB/c nude mice	↑ migration, invasion, EMT process, viability, and stemness	[167]
Gastric cancer	PART1	AR interacts with PART1 to stimulatePLZF expression	PLZF, EZH2, PDGFRβ/PI3K/Akt signaling pathway	GES-1, MGC-803, BGC-823, and SGC-7901/GEOdatabase: GSE27342, GSE33335, andGSE3072, plus 136 GC samples	↓ migration, invasion, and metastasis	[168]
Clear cell renal cell carcinoma	TANAR	↑	TWIST1	786O, SW839, HEK293T/51 ccRCC tissues, and 23 ANCTs/male athymic BALB/c nude mice	↑ VM formation	[169]
	circHIAT1	↓	HIAT1, miR-195-5p/29a-3p/29c-3p, and CDC42	SRC-2, VHL(þ) Caki-1, SW-839, and ACHN	↓ migration and invasion	[170]
	circEXOC7	↓	DHX9, miR-149-3p, CSF1	SW839, 786-O, Caki-1, ACHN, HEK293T/4 RCC samples with RBM, and 10 RCC samples without RBM/Balb/c nude mice	↑ RBM and osteolytic formation	[171]

## Data Availability

Not applicable.

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
