# Peer review of "Interaction between Non-Coding RNAs and Androgen Receptor with an Especial Focus on Prostate Cancer"

_cells, 2021, doi:10.3390/cells10113198_

Round 1
Reviewer 1 Report
In current review manuscript, the authors narrated the non-coding RNAs which includes sRNAs, lncRNAs and circular RNAs and their involvement with the Androgen Receptor (AR). The review is comprehensive, begins with the narration on Androgen receptor structure and its role in modulating ncRNA expression and their effects on Androgen Receptor. The style and English is well and connected not only about prostate cancer but also explains other cancer types such as bladder cancer, Melanoma, Renal carcinoma, and Esophageal squamous cell carcinoma. The comprehensive review article will further help to understand and design the strategies for effective therapeutic drugs.
Author Response
Thanks for your encouraging comments.
Reviewer 2 Report
AR plays a key role in prostate cancer. In recent years, there have been many research papers on ncRNA related to AR. Therefore, it is necessary to review these results. The article cited a large number of references with a lot of content, which is very valuable, but I think it still needs improvement in certain aspects. The article is more likely to be a simple statement of the results of others, and the organization is not clear enough, and the summary is not enough. Please see the revised comments below:
- Summarize several different ways that ncRNAs affect AR
- Summarize the mechanism of AR on ncRNA (AR as a transcription factor directly or indirectly regulates the transcription of ncRNAs)
- It is recommended to add subtitles under the headline, for example (for reference only):
- Effects of miRNAs on AR
(1)Acting on AR mRNA to directly negatively regulate AR expression
(2)Indirectly regulate AR expression or AR signal
(3)Effects of miRNAs on AR in other different cancer types
- Regulatory impact of lncRNAs on AR
(1)regulation of AR expression
(2)regulation of AR activity as AR-interacting partner
(3)Effects of miRNAs on AR in other different cancer types
- Effects of AR on ncRNAs
(1)AR responsive miRNA
(2)AR responsive lncRNA and circRNA
In addition, there are some minor revisions:
- ref. 6 and ref. 7 were repeated.
- In the description of Table 1 , BPH was repeated twice.
- I suggest to cite this newly published article: Comprehensive Characterization of Androgen-Responsive circRNAs in Prostate Cancer. Life (Basel), 2021 Oct 15;11(10):1096.
Author Response
We summarized the mentioned points.
We added subtitles.
We edited the mentioned minor points.
Round 2
Reviewer 2 Report
None